# PIMGAVir and Vir-MinION: Two Viral Metagenomic Pipelines for Complete Baseline Analysis of 2nd and 3rd Generation Data

**DOI:** 10.3390/v14061260

**Published:** 2022-06-10

**Authors:** Emilio Mastriani, Kathrina Mae Bienes, Gary Wong, Nicolas Berthet

**Affiliations:** 1Unit of Discovery and Molecular Characterization of Pathogens, Centre for Microbes, Development and Health, Institut Pasteur of Shanghai, Chinese Academy of Sciences, Shanghai 200031, China; emiliomastrani@icloud.com (E.M.); kathrina@ips.ac.cn (K.M.B.); 2Viral Hemorrhagic Fevers Research Unit, Institut Pasteur of Shanghai, Chinese Academy of Sciences, Shanghai 200031, China; 3Unité Environnement et Risque Infectieux, Institut Pasteur, Cellule d’Intervention Biologique d’Urgence, 75015 Paris, France

**Keywords:** taxonomic classification, metagenomic pipeline, 2nd and 3rd sequencing generation, multiple strategies analysis

## Abstract

The taxonomic classification of viral sequences is frequently used for the rapid identification of pathogens, which is a key point for when a viral outbreak occurs. Both Oxford Nanopore Technologies (ONT) MinION and the Illumina (NGS) technology provide efficient methods to detect viral pathogens. Despite the availability of many strategies and software, matching them can be a very tedious and time-consuming task. As a result, we developed PIMGAVir and Vir-MinION, two metagenomics pipelines that automatically provide the user with a complete baseline analysis. The PIMGAVir and Vir-MinION pipelines work on 2nd and 3rd generation data, respectively, and provide the user with a taxonomic classification of the reads through three strategies: assembly-based, read-based, and clustering-based. The pipelines supply the scientist with comprehensive results in graphical and textual format for future analyses. Finally, the pipelines equip the user with a stand-alone platform with dedicated and various viral databases, which is a requirement for working in field conditions without internet connection.

## 1. Introduction

Recent advances in next-generation sequencing (NGS) technologies and computational methods are revolutionizing scientific research in public health [1]. One such application of NGS is metagenomics. Metagenomic sequencing (mNGS) is an unbiased, culture-independent approach that analyzes the nucleic acid content of any clinical or environmental sample [2,3,4,5]. Clinical metagenomics (CMg) is a method of choice for detecting and identifying infectious etiologies [6,7]. In addition to pathogen detection and identification, infectious disease surveillance also provides information on disease transmission, strain type, virulence profile, antimicrobial susceptibility, and other information relevant to outbreak investigation and treatment guidelines [3,7,8,9,10].

Considering that most emerging infectious diseases (EIDs) in humans originate from wildlife known to harbor many zoonotic pathogens, such as bats, NGS are now considered essential tools for the molecular characterization of viral communities that could help determine the origin of outbreaks and discover new pathogens [10,11]. Due to the incessant emergence of zoonotic diseases, a constant search for emerging infectious etiological agents is deemed necessary. Two main approaches are currently available for the search of these infectious agents by NGS, either using short (e.g., with the 2nd generation technology with sequencers marketed by Illumina) or long (e.g., with the 3rd generation technology with sequencers marketed by Oxford Nanopore Technologies) reads. Beyond the difference in read size, the use of 2G (Illumina) allows for a much higher coverage and sequencing depth but can take 20 to 60 h, whereas sequencing based on nanopore sequencing technology allows a direct and real-time availability of sequencing data and a reduction of sequencing time from several days to only a few hours. Although the sequencing depth is less than that of 2G technology, 3G technology has tremendous potential in clinical sequencing applications at the point of care, whether at the bedside or in the field, due to its portability, speed, flexibility and relatively low cost of the device [12,13,14]. However, whatever sequencing technology is used for virological investigation, it requires the development and use of dedicated workflows for processing these data. Several workflows for the analyses of viral metagenomic data obtained from 2nd and 3rd generation sequencers exist in the literature [15,16,17]. These can be distinguished to perform either time-constrained diagnostics, surveillance and monitoring of epidemics, remote homology detection (discovery), and biodiversity studies [18].

In the field of metagenomics for pathogen research, workflows are divided into three main areas that make up the methodological approaches of workflows: read-based approaches, assembly-based approaches, and clustering-based approaches. The first read-based strategy analyzes unassembled short reads to identify the overall taxonomic/functional composition of samples. The usual main steps of this approach are: reading the QC [19], merging the reads [20], mapping to the NR for taxonomic data [21,22], and analyzing the summaries of the taxonomic and functional distributions. The MG-RAST server is a most representative example of this approach based on short read analysis [23]. The second strategy that is based on assembly attempts to assemble reads from one or more samples, and “classify” the contigs from these samples into genomes to analyze genes and contigs. It identifies the functional and metabolic capabilities of specific microbes in the samples. As before, the workflow includes classical steps such as quality control of reads [19] and read merging, but there are additional assembly steps [24,25], such as mapping of reads from each sample to contigs for quantification and clustering [26]. The genome clustering, contig composition, and mapping data are used to group contigs into “genomic bins” [27,28], eventually moving to de novo gene annotation [29,30] and performing gene annotation in read-centric approaches. The IMG from JGI is an example of a workflow that is based on a short read assembly-centric strategy [31]. The last approach, which is based on clustering, includes the same steps of quality control, merging, and filtering as previously described, but there is an additional clustering step [32,33]. This last step results in “centroid” sequences, i.e., sequences representative of each group, which are transmitted downstream of the pipeline. The VirIdAl box is an example of a workflow based on a clustering approach [34].

Although there are differences that are required for the analyses of long reads versus short reads, the analytical approaches are divided into the same three clusters. Regardless of the chosen analytical approach, the preprocessing steps always start with the steps of base calling [35], demultiplexing [35,36], filtering and quality control [37]. The read-only strategy relies on the identification of reads by taxa using an algorithm with Centrifuge software [38] and the NCBI RefSeq sequence database, as is the case with the Metrichor/EPI2ME cloud platform (Metrichor Ltd., Oxford, UK). Strategies based on assembly and clustering share, with the previous strategy, the pre-processing steps to which the assembly [39,40], polishing [41,42,43], or clustering [32,44] steps must be added as described in the MicroPIPE [45], NanoCLUST [46], and mothur [47] workflows.

Nowadays, it is very common in a research laboratory to combine the different sequencing technologies available to perform metagenomic studies. Therefore, it is normal to use different pipelines, based on a specific strategy, to perform taxonomic classifications of large amounts of sequencing data depending on the strategy adopted. As we are not aware of any bioinformatics pipeline that can combine the three analysis strategies in a single workflow, the establishment of a connection between the different existing workflows may require time and qualified human resources due to the problems that may be encountered stemming from the lack of compatibility between various workflows. In this technical note, we present PIMGAVir (PIpeline for MetaGenomic Analysis of Viruses) and Vir-MinION (Viral MinION pipeline), which are two viral metagenomic pipelines designed to provide scientists with a complete baseline analysis of viral sequences from 2G and 3G sequencers. Both pipelines are freely downloadable (Appendix A) and allow the user to run one or more of the three approaches independently.

## 2. Objectives of PIMGAVir and Vir-MinION Pipelines

The main objectives of the PIMGAVir and Vir-MinION pipelines are to provide a complete taxonomic classification basis for reads from either 2G or 3G sequencing, respectively, to hide the complexity of the process from the end user, to save runtime and to be usable in offline conditions. They have been designed to automatically perform both pre-processing and contaminant removal tasks of a defined host and bacterial genomes as well as to execute one or more strategies in parallel to perform taxonomic classification, thus obtaining a wide range of results. Finally, PIMGAVir and Vir-MinION all present results in text/tabular and graphical plots.

### 2.1. Description of the Two Pipelines

Given that the objectives of these two pipelines are similar, they share the use of identical computer packages as for example with the use of megahit for the assembly step (Table 1). However, they also differ in the use of dedicated packages (Table 1), such as the use of filtering or demultiplexing tools that are specific to 3G data processing. On the other hand, both pipelines use the same database that will be used for the filtering steps of the contaminants, e.g., Silva to remove bacterial sequences or NR refseq for the identification of viral sequences (Table 2).

As shown in Figure 1A, the pipeline executes the pre-processing task to trim the raw data and remove contaminants. Then, according to the user option, the reads_filtering script will filter out the reads not belonging to desired taxa. At this point, the pipeline will execute one or more strategies (namely, read_based, ass_based, and clust_based) in parallel to proceed with the taxonomic classification. Double applications perform both the clustering and assembly methods to present the user with a pool of comparable results. The pipeline builds a specific data structure following the logical schema “strategy-application” to be easily surfable. For example, Figure 1B depicts the data structure created during the analysis step. The PIMGAVir pipeline uses a set of local-viral databases to perform both the filtering and taxonomic tasks (Figure 1C). The pipeline runs under the Ubuntu 20.04 operative system, and a set of bash scripts performs the workflow. Each strategy, once called, executes a few scripts and produces a collection of results (text, HTML, and pdf) and log files (Figure 2A). Most of the scripts lean on a group of applications and databases to accomplish their task. Figure 2B shows the databases and applications used by every script. Finally, the user has the freedom to run every one of the mentioned scripts as an autonomous process as long as the input format is respected. The three strategies have been designed to run independently, allowing the user to run on parallel computing systems. The PIMGAVir pipeline has been tested on a cluster configured on the SLURM workload manager, running on multiple samples at once. The following is an example of the SLURM script to run the PIMGAVir pipeline (Figure 2C).

Regarding the Vir-MinION pipeline, after the pre-processing step, which is executed as the default step, the pipeline runs one or more methods in parallel, according to the user choice. The read_based strategy carries out the taxonomic classification using the demultiplexing results as input to generate an overall view of what the sample contains. The clust_based approach, as the name suggests, identifies the clusters obtained from the meta-barcoding step and executes the taxonomic classification on them. In the ass_based mode, the pipeline performs the assembly step from the shotgun, producing their taxonomic classification. As in the same case of PIMGAVir, the Vir-MinION pipeline relies on local-viral DBs to guarantee its capability in connection-less conditions and save runtime. The use of both taxonomic classifiers (Kraken2 and Kaiju) gives the user the possibility to compare the results. The outcomes are presented in graphical and text tabular layouts for further analysis. The Vir-MinION pipeline runs under Ubuntu 20.04 and uses NVIDIA technology to make its processing. The Vir-MinION utilizes a collection of bash scripts to perform the workflow. For its part, the bash scripts invoke a group of applications and databases to accomplish their task. Figure 3B shows the databases and applications used by every script. As shown in Figure 3C, the pipeline builds a specific data structure following the logical schema “strategy-application” to be easily surfable.

### 2.2. Test Pipelines

Validation of both pipelines was performed using simulated data using CAMISIM [59] to generate unique communities for 2G and 3G data. DeepSignal [60] was used to simulate the MinION signal from the already available community. The community consists of a bacterial genome, *Helicobacter hepaticus* ATCC 51449, and two viral RNA genomes, *Hepatitis A virus* (HAV) (NC_007905.1) and *Ippy virus*, which consists of two *S* and *L* segments (NC_007906.1). The distribution of the reads is described in Table 3. Table 4 presents the data flow followed by PIMGAVir, starting from the simulated data to the data treated with the different approaches, ready to be classified. The percentage associated to the ribosomal removing and to the filtration step of the non-viral reads of PIMGAVir showed that a low ratio of ribosomal contaminants has been removed, while the reads corresponding to *Helicobacter hepaticus* were correctly discarded in large part and that most of the reads corresponding to the viral genomes were used as input for the classification step with different tools (Kraken, Kaiju and BLASTN) (Table 5). At the same time, a high percentage of reads has been discarded during the assembly/clustering step, showing how much the pipeline is sensible to the automatic improvement of the draft assemblies (performed by PILON) and to the de-replication and chimera removing (performed by vsearch), during the assembly and clustering step, respectively. An average percentage of reads of 0.30% (0.07% to 0.55%) and 0.33% (0.11% to 0.55%), respectively, for *HAV* and *Ippy virus*, were correctly classified. Indeed, the percentage of coverage (mapping analysis based on reference sequences) detains the average value of 92% for *HAV* and the values of 97.25% and 47.3% for Ippy S and L, respectively. The high score of accuracy calculated as the percentage of alignment with the reference genome supports the correctness of the reads’ classification. A retrospective analysis of the unclassified reads shows that they corresponded to 96.5%. Both after the clustering steps and by the two different assembly approaches, the number of clusters and contigs is lower than the number of initial reads (4 to 247, for contigs and clusters, respectively). However, the coverage of both genomes is bigger than 72% and 90%, respectively, for the *Ippy* and *HAV* genomes.

## 3. Discussion and Conclusions

PIMGAVir and Vir-MinION are free, connection-less, and modular automated metagenomics pipelines that provide the user with a complete baseline analysis for the taxonomic classification of the reads. The PIMGAVir pipeline works on data from the 2nd generation technology, while Vir-MinION works on 3rd generation technology. We designed the applications to be easily used by biologists and generally by users without particular computer skills. Although the pipelines do not have a graphical or web interface, both of them need only a few command line parameters. The required parameters are easy to understand, such as the input files to be analyzed, the strategy to carry out the analysis, or the number of cores you would like to allocate. We tested the pipelines on the Desktop equipped with i9-12900KF as CPU, 64 GB DDR5 of RAM, and GeForce Nvidia 3080Ti with 12 GB of RAM. The PIMGAVir pipeline required about 14 h of execution to generate the results from all the three approaches with an input of coupled fastq files of six million reads per file, while the Vir-MinION pipeline took two hours to complete the three strategies, using an input of 94 GB of fast5 files from 12 barcodes with a total of four million long reads. The short run time for the Vir-MinION pipeline emphasizes its utility as a valuable support for field applications, such as “quasi-real-time” pandemic monitoring.

The PIMGAVir and Vir-MinION pipelines, which approach metagenomic analysis from these three different angles, will provide the user with a potentially complementary set of information, as each approach will answer specific questions. Indeed, metagenomics based on unassembled reads is valuable for quantitative analysis, while assembly-based workflows will be used to identify the different organisms residing within the samples. The assembly-based strategy groups metagenomic contigs into potential genomes to study the functional roles of microbial populations. Thus, the combined analysis of these results can help to better define the most plausible viral metagenomic composition of samples. In addition, the adoption of multi-solution software specific to viral genome analysis has increased the reliability and computational efficiency of these pipelines where possible. For example, the choice of the assembler is fundamental before executing the taxonomic classification, and many software progams/algorithms exist to perform this task. Moreover, when working with a new dataset, it is common to generate a few assemblies testing different programs with different parameters, to compare the results and thus be more confident we are doing the best with the data. From this perspective, SPAdes and MEGAHIT are the two most commonly used assemblers today. SPAdes uses much more memory than MEGAHIT, so it is often more suitable for working with one or a few genomes (such as from an isolate or enrichment culture). However, if working with high-diversity metagenomic samples, sometimes, the memory requirements for SPAdes are too high, and MEGAHIT (which uses much less memory) can handle the task instead. Since the PIMGAVir pipeline uses both assemblers to produce the assembled genomes, the Vir-MinION pipeline (upon the same philosophy) accomplishes the assembly steps using either MEGAHIT and Flye assemblers. Of course, the consistency of the databases is also a crucial point during the taxonomic classification, and we have chosen to classify every “object” (whether reads, clusters, or assembled genome) with different software (kraken2, kaiju, or blastn) querying to several viral databases.

Each pipeline has to continue to evolve through further studies of comparison with other current or new pipelines or other new tools that will be developed in the future, as exemplified by VIBRANT [61] and VirSorter [62]. Another point is to phase out concerns to the running time needed by the PIMGAVir pipeline. As mentioned before, the PIMGAVir pipeline has been tested on a small cluster of seven worker nodes, communicating over Ethernet and equipped with shared remote storage. The cluster was configured on a SLURM workload manager with shared user’s home and password-less access. Being the DBs instantiated on the remote storage, the queries over NFS required a relevant amount of running time. Further investigation can be performed to optimize the communication between the processes and the DB’s performances.

In conclusion, these two pipelines, PIMGAVir and Vir-MinION, have already been used in our laboratory for the search and identification of known or new pathogens from meta-transcriptomic data obtained from a wide variety of hosts such as bats, arthropods, ectoparasites or wild and domestic rodents. However, they can be used by many other researchers, whose applications require metagenomic classification of their 2nd or 3rd generation data.

## Figures and Tables

**Figure 1 viruses-14-01260-f001:**
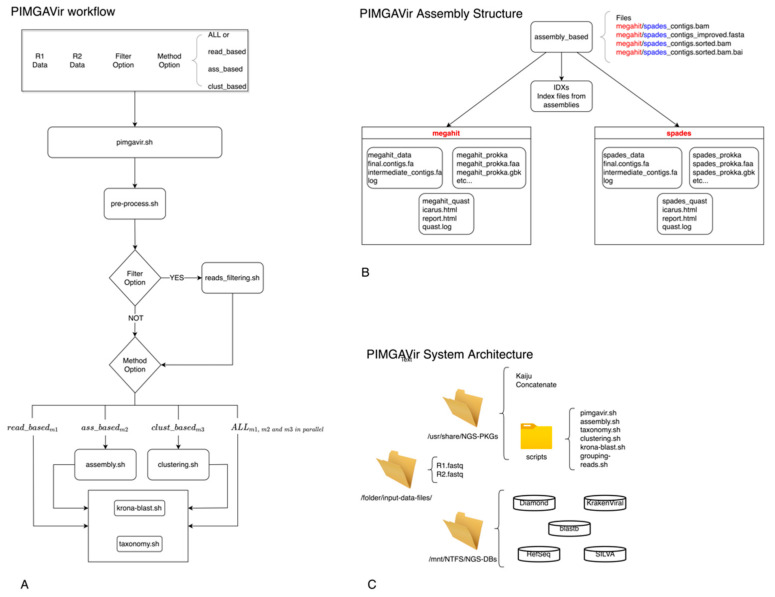
PIMGAVir pipeline. (**A**) PIMGAVir flowchart showing the pre-processing task, the filtering option, the execution of the three strategies and the taxonomic classification. (**B**) Example of the data structure organization for the ass_based strategy. (**C**) Simplification of the PIMGAVir system architecture.

**Figure 2 viruses-14-01260-f002:**
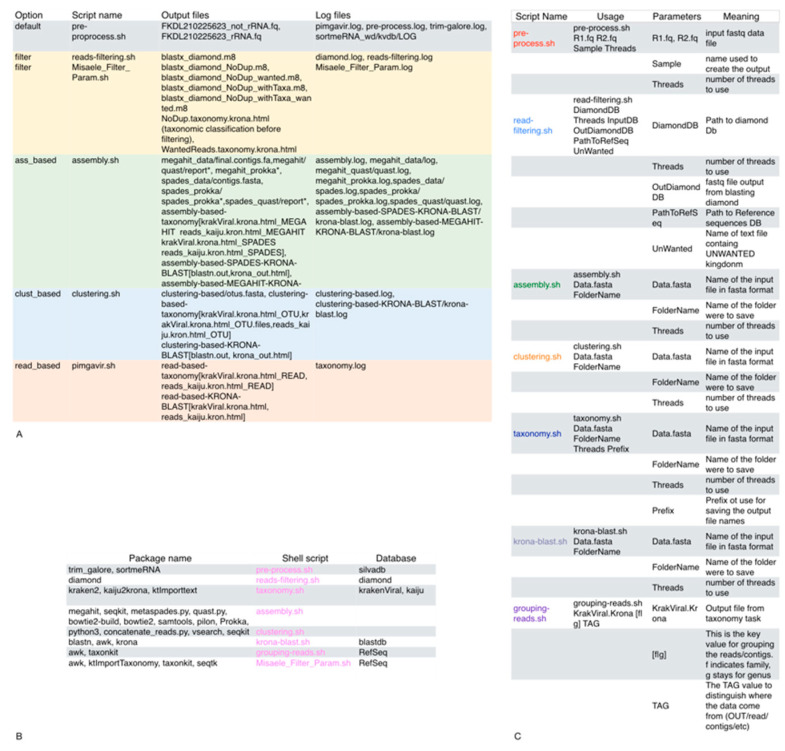
PIMGAVir scripts. (**A**) Table showing some of the results and log files produced by every strategy and the invoked script. (**B**) Table reporting the databases and the packages used by every script. (**C**) Table containing the list of scripts used in PIMGAVir and its parameters.

**Figure 3 viruses-14-01260-f003:**
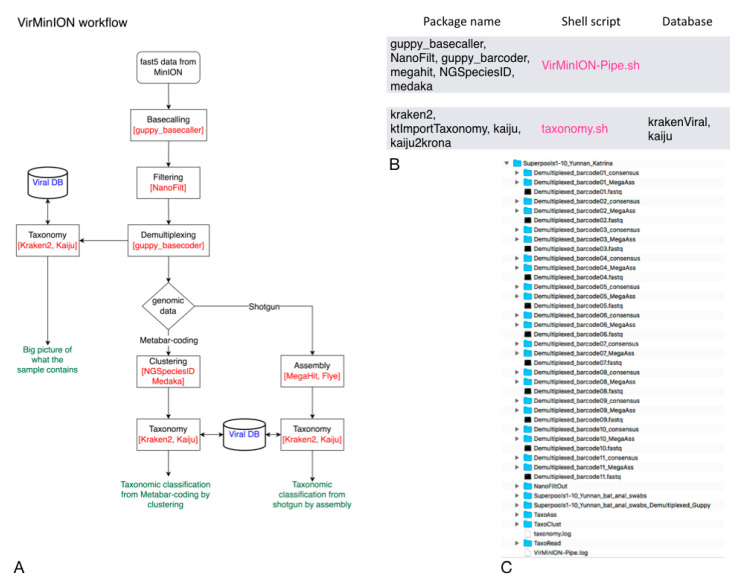
Vir-MinION pipeline. (**A**) Vir-MinION flowchart showing the execution of the three strategies. (**B**) Vir-MinION table showing the scripts, packages and DBs used. (**C**) Vir-MinION data structure at the end of the ALL strategy execution.

**Table 1 viruses-14-01260-t001:** List of used packages from PIMGAVir and Vir-MinION.

Package Name	Version	Pipeline	Task
TrimGalore [48]	0.6.5-1	PIMGAVir	preprocessing
SortMeRNA [49]	4.3.4	PIMGAVir	filtering
diamond [22]	2.0.11.149	PIMGAVir	filtering
KronaTools [50]	2.8.1	PIMGAVir	filtering
Taxonkit [51]	0.10.1	PIMGAVir	filtering
seqtk [52]	1.3	PIMGAVir	filtering
megahit [24]	v1.2.9	PIMGAVir/Vir-MinION	assembly
flye [40]	v2.9	Vir-MinION	assembly
quast [53]	v5.0.2	PIMGAVir	assembly
spades [25]	3.13.1	PIMGAVir	assembly
bowtie2 [26]	2.4.4	PIMGAVir	assembly
samtools [54]	1.10-3	PIMGAVir	assembly
pilon [42]	1.23	PIMGAVir	assembly
Prokka [30]	1.14.6	PIMGAVir	assembly
kraken2 [55]	2.1.2	PIMGAVir/Vir-MinION	taxonomy
kaiju [56]	1.8.2	PIMGAVir/Vir-MinION	taxonomy
blastn [21]	2.9.0+	PIMGAVir/Vir-MinION	taxonomy
seqkit [57]	2.0.0	PIMGAVir	clustering
vsearch [32]	v2.18.0	PIMGAVir	clustering
guppy_basecaller [35]	5.0.13	Vir-MinION	basecalling
NanoFilt [58]	2.3.0	Vir-MinION	filtering
guppy_barcoder [35]	5.0.13	Vir-MinION	demultiplexing
NGSpeciesID [44]	0.1.2.1	Vir-MinION	clustering
medaka [43]	0.11.5	Vir-MinION	clustering

**Table 2 viruses-14-01260-t002:** Databases used by PIMGAVir and Vir-MinION.

Database Name	Version	Build Up Date
diamond/refseq_protein_nonredund	Refseq protein non redundant genomes Database format version = 3	6–14 May 2022
krakendb/SILVA_138.1_SSURef_NR99_tax_silva	Silva DB v. 138.1	17 May 2021
krakenviral/database.kraken	Kraken Viral DB v2.0.8	17 May 2021
NCBI-RefSeq/viralseq_2021-12-14_14-45-53	Refseq viruses’ representative genomes. BLASTDB Version: 5	14 December 2021
SILVA/ssr138, slr138	Ribosomal DB for SSR138 and SLR138	27 August 2020

**Table 3 viruses-14-01260-t003:** Statistical values. The table reports the statistical values of the simulated data produced using CAMISIM.

		Reference Genome	Fragment Mean Size	Total Number of Reads	Number of Reads per Genome	Coverage with the Reference (%)	Average Depth per Genome
**Simulated data type**	2G	*Ippy virus segment S*	270	666,618	800	0.12	34.31
		*Ippy virus segment L*			666	0.01	1.35
		*Hepatitis A virus*			733	0.11	15.12
		*Helicobacter hepaticus*			655,218	98.29	55.44
	3G	*Ippy virus segment S*	500	13,499	5382	0.20	3348.52
		*Ippy virus segment L*			423	0.03	198.974
		*Hepatitis A virus*			5485	0.18	2571.08
		*Helicobacter hepaticus*			2205	97.30	5.07815

**Table 4 viruses-14-01260-t004:** PIMGAVir, numerical data flow of data during its execution.

Task	Starting Number of Reads	Ending Number of Reads	Removed (Number)	Removed (Percentage)
Trimming	666,618	642,614	24254	3.64
Ribosomal removing	642,614	640,614	1750	0.26
Filtering unwanted	640,614	170,960	469,654	70.45
MEGAHIT	170,960	884	170,086	99.48
SPADES	170,960	1354	169,616	99.20
clustering	170,960	44,783	126,187	73.80

**Table 5 viruses-14-01260-t005:** PIMGAVir statistics. The table shows the statistical values associated to the taxonomic classification of PIMGAVir.

Approach	DB			Hepatitis A Virus—taxid: 12092				Ippy Virus (Segment S + L)—taxid: 55096					
	Name	Total of Reads/Cluster/Contigs Analyzed by the DB	Average Size (bp per Read)	Total of Reads/Cluster/Contigs Classified by the DB	% of Reads/Contigs Collected	Coverage of Genome (%)	Accuracy % of Align	Total of Reads/Contigs Classified by the DB	% of Reads/Contigs Collected	S (Ippy-01)		L (Ippy-02)	
										% of Coverage	Accuracy %of Align	% of Coverage	Accuracy %of Align
read-based	kraken	170,960	139.9	648	0.38	91	100	790	0.46	98	91.65	69	7.97
	Kaiju	170,960	139.9	644	0.37	91	38.69	780	0.45	98	44.42	72	3.92
	BLASTN	170,690	139.9	648	0.37	100	100	793	0.46	98	91.42	72	8.07
clustering-based	kraken	44,783	145.7	247	0.55	100	100	237	0.55	97	76.25	70	22.50
	kaiju	44,783	145.7	244	0.54	90	100	233	0.52	97	75.97	72	22.75
	BLASTN	44,783	145.7	247	0.55	90	100	241	0.54	97	76.25	72	22.50
assembly-based-megahit	kraken	884	728.6	4	0.45	90	100	1	0.11	96	100	0	0
	kaiju	884	728.6	4	0.45	90	100	1	0.11	96	100	0	0
	BLASTN	884	728.6	4	0.45	90	100	1	0.11	96	100	0	0
assembly-based-spades	kraken	1354	617	1	0.07	91	100	7	0.52	98	14.29	23	85.71
	kaiju	1354	617	1	0.07	91	100	7	0.52	98	14.29	23	85.71
	BLASTN	1354	617	1	0.07	91	100	7	0.52	98	14.29	23	85.71

## Data Availability

Not applicable.

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
