# Peer review of "PIMGAVir and Vir-MinION: Two Viral Metagenomic Pipelines for Complete Baseline Analysis of 2nd and 3rd Generation Data"

_viruses, 2022, doi:10.3390/v14061260_

Round 1
Reviewer 1 Report
The authors developed two pipelines for metagenomic viral analysis, one for short-read sequencing and the other for long-read sequencing, in which included reads-, assembly- and cluster-based strategies. These pipelines are valuable to biologists with less bioinformatics experience and will be of very interest to readers in the field of viral study. While 3G metagenomics pipeline took about 2 hours to complete, the 2G one took about 14 hours, which seems too long for me, personally. The authors may consider parallel computing for three approaches (in discussion, at least) so as to save time in data processing, though it depends on the computer being used.
Additional concerns are shown as below:
1) Databases are key to the analysis and are time-sensitive. Multiple databases from various sources were used for differenct approaches. It will be much better to list the build-up date of each database, which may be useful for interpretation of results difference between approaches, if any existing. Updating databases in time is also important, while more viruses are being identified each day.
2) It was great for the authors giving the details of algorithms installation and setup. It will be much greater if the authors could provide a couple sets of raw data as examples for readers to go through these two pipelines, respectively, and three approaches, results from which will definitely help the demonstration.
3) Fonts in figures are too small to be viewed. Certain improvement is needed.
Overall, the manuscript is well-written, and I enjoyed reading.
Author Response
1. Even if some tasks have to be executed in a serialized way (e.g. pre-processing, filtering, ...) the three strategies have been thought to run independently, almost ready to run on parallel computing systems. Because of the difficulty to have a cluster equipped with NVidia/CUDA interface, the Vir-MiION pipeline has been tested only on Alienware WorkStation, while PIMGAVir ran successfully on parallel computing infrastructure. In particular, the PIMGAVir pipeline has been tested on a small cluster of 7 worker nodes, communicating over Ethernet and equipped with shared remote storage. The cluster was configured on SLURM workload manager with shared user's home and password-less access. Being the DBs instantiated on the remote storage, we observed the queries over NFS requiring a relevant amount of running time. Due to this reason, we opted to use the pipeline on a speedy workstation. The topic has been mentioned in the discussion section too.
2-3. DBS Build-date: Growing very quickly the number of newly discovered viruses, we agree with the reviewer about the importance of updating the DBs used in the pipelines. We inserted a new column into Table 2 indicating the BuildUp-Date of each used database. We also published a new document on https://github.com/emiliomastriani/PIMGAVir/blob/main/README.md to drive the user during the DBs installation.
4. According to the reviewer comment, two additional files have been published, the VIRMINION-CAMISIM_DEEPSIM and PIMGAVIR-CAMISIM documents. The two guides, available at the project repository at https://github.com/emiliomastriani/, will drive the user on build-up the desired viral population in a "synthetic" way and check how both the pipelines work. The synthetic data we used for testing the pipelines have been published at the relative project repository too.
5. Figures with higher quality have been produced
Reviewer 2 Report
Dear authors,
you submitted a high-quality paper that will be useful for scientists working in virology and that can be also applied on site. I just ask you to read carefully the manuscript and add appropriate references, especially in the discussion (cf file attached).
Best,
the reviewer

Author Response
1. Line 141, changed de-novo to italics
2. Lines 202-217, the following softwares are mentioned: SPAdes, MEGAHIT, Flye. They are already cited in Table 1
Reviewer 3 Report
The presented work is devoted to the actual topic, the development of a pipeline for the analysis of data from high-throughput sequencing of viral communities in order to establish the taxonomic diversity of viruses. The widespread introduction of such pipelines would allow a large number of biologists, virologists and physicians who are not experts in the field of data analysis to use metagenomic analysis for the study of viral communities. The text of the manuscript is well structured, the objectives of the study and conclusions are clearly formulated.
A number of major comment s can be made on the work:
- In table 1, the «Task» column should give a more detailed description of the analysis stage at which this program will be applied.
- Most of the programs used, such as "diamond", "kraken2", "spades", "bowtie2", "blastn" and others, have a whole set of parameters that allow you to adjust the tasks performed. What parameter values of these and other programs are used in the analysis? Can the user of the pipeline adjust the values of these parameters himself? For example, the "diamond" program, when comparing reads with protein databases, can translate nucleotide sequences both according to the standard and according to the bacterial genetic code. This is important for what we want to filter out of the prokaryote or eukaryote dataset. It is necessary to add a column to the table 1 indicating the parameters of operation of all the indicated programs.
- In table 2, you need to add a column indicating the parts of the pipeline and the programs that use these databases.
- I did not find the possibility of comparative analysis of several samples at the same time both in the text of the work itself and on the pipeline site at "https://github.com/". Such a possibility would allow researchers to obtain immediately, without additional manipulations, a table of the representation of viral taxa in different samples (table row - virus taxon, table columns - the number of reads per this taxon in different samples). Such a table would simplify analysis for less experienced users.
- I think it's a very bad idea to pre-filter the set of reads from non-viral DNA before assembling in an «ass_based» pipeline. Carrying out such filtering, we can take for viruses prophages and proviruses encountered in the genome of prokaryotes and eukaryotes. The collection of all data in its entirety, and then taxonomic analysis, would make it possible to single out scaffolds of prokaryotes and eukaryotes containing prophages and proviruses as a separate category of viral elements. Some work (Coutinho, et al 2020) that did a complete assembling of all data with isolating prophages in prokaryotic genomes. In addition, by filtering data from bacterial and eukaryotic DNA, we can lose metabolic genes obtained from their hosts (AMG - auxiliary metabolic genes) in viral scaffolds.
- Figures 1, 2 and 3 contain very small, hard to read text. You need to increase the font size on these pictures and increase the resolution to 300-600 dpi.
- There is no comparison developed by the authors of the «PIMGAVir» and «Vir-MinION» pipelines with other pipelines for processing virus community analysis data. I recommend comparing the «ass_based» pipeline for the number of scaffolds identified as viral with the "virsorter" (Roux, et al 2015) and "VIBRANT" (Kieft, et al 2020) pipelines.
Coutinho, F. H., Cabello-Yeves, P. J., Gonzalez-Serrano, R., Rosselli, R., López-Pérez, M., Zemskaya, T. I., ... & Rodriguez-Valera, F. (2020). New viral biogeochemical roles revealed through metagenomic analysis of Lake Baikal. Microbiome, 8(1), 1-15.
Kieft, K., Zhou, Z., & Anantharaman, K. (2020). VIBRANT: automated recovery, annotation and curation of microbial viruses, and evaluation of viral community function from genomic sequences. Microbiome, 8(1), 1-23.
Roux, S., Enault, F., Hurwitz, B. L., & Sullivan, M. B. (2015). VirSorter: mining viral signal from microbial genomic data. PeerJ, 3, e985.
After eliminating the comments, doing additional work on data analysis and re-reviewing the manuscripts, it can be published in the journal.
Author Response
1. The information reported in Figure 2 and Table 1 are complementary. The reader can obtain more information jointly consulting the Table 1 and Figure 2 A.
2. Both the pipelines are focused on the taxonomic classification of viral reads, so we configured all the programs to this scope. Being the programs of both the pipelines handled by bash scripts, the user is free to change all the parameters, according to the needs. The User Manuals of both the pipelines have been updated [https://github.com/emiliomastriani]. A supplementary table has been added reporting relevant parameters used during the script’s execution and its relative program. The user can refer to it to customize the pipeline, according to the data/experiment.
3. The information reported in Figure 2 and Table 2 are complementary. The reader can obtain more information jointly consulting the Table 2 and Figure 2 B.
4. The three strategies have been thought to run independently, allowing the user to run on parallel computing systems. The PIMGAVir pipeline has been tested on a cluster con-figured on SLURM workload manager, running on multiple samples at once. The discussion has been mentioned in the manuscript and one example on how-to-submit a pimgavir job has been inserted in the manuscript
5. The application of the filtering task (--filter) is optional, as mentioned in the MS ("according to the user option, the reads_filtering script will filter", line 133)
6. Thank you for the suggestions. The comparison with already existing pipelines will be one of the next steps. The topic has been mentioned in the discussion section.
7. Figures with higher quality have been produced
Reviewer 4 Report
The manuscript entitled “PIMGAVir and Vir-MinION: two viral metagenomic pipelines for complete baseline analysis of 2nd and 3rd generation data” is a good manuscript. Authors have developed two automatic metagenomics pipelines namely, PIMGAVir and Vir-MinION. These pipelines are applicable to 2nd and 3rd generation data.
I hope these developed pipelines may be useful in future for taxonomic classification of the reads.
It is a well-written manuscript.
Introduction has been written very well. Methods are well explained
Results and figures are well explained.
Discussion and conclusion explain very well the outcomes of this manuscript.
Author Response
Thank you
Reviewer 5 Report
The study by Mastriani et al. developed two practical pipelines for viral metagenomics using available bioinformatical tools. The overall design seems fine, and this reviewer would ask a significant question is the authors test the two pipelines using the above pipelines with either simulated or actual datasets? Without a series of results to prove their performance and accuracy, it is challenging to draw convincible conclusions. One minor issue is the data presentation. The authors should use a larger font size in all figures and label each panel (A, B, C...) in the left corner.
Author Response
We have tested both pipelines using simulated data. We created a unique community of four complete viral genomes: AE017125.1 Helicobacter hepaticus ATCC 51449, NC_001489.1 Hepatitis A virus, NC_007905.1 Ippy virus segment S, and NC_007906.1 Ippy virus segment L. We used CAMISIM [insert reference] to create the viral community for 2G and 3G data and we referred to DeepSignal [insert reference] to simulate the MinION signal starting from the already available viral community. The taxonomic profile of the sample used for the test reported the following percentages: Hepatovirus A (14.36%), Ippy Mammarena (33.9%), Helicobacter (51.69%), while the Coverage with the reference genomes (%) reported the following distribution:
Ippy S --> 0.12
Ippy L --> 0.01
Hepatitis A --> 0.11
Helicobacter hepaticus --> 98.29
Results obtained at the end of PIMGAVir execution, show the Helicobacter has been correctly discarded (being interested only in viral population), and the viral genomes identified with assembly-SPADES and read-based approaches performing better. We gained similar results by the Vir-MinION pipeline, using the same virus profile as input. Data show the Helicobacter has been dropped out, while the read-based approach coupled with the viral DBs gives back the viral proportion closest to the simulated data. In both cases, the pipelines correctly identified the viral composition of the sample, even if further tests from samples with different viral genomes or distributions and a more accurate calculation of the viruses' abundances will help to improve the pipelines’ performance. Please, refer to the "Test pipelines" paragraph in the manuscript.
At the end, figures with higher quality have been produced
Round 2
Reviewer 3 Report
I reviewed the new version of the manuscript and the responses to the comments made. Presentation of materials and justification of the results have improved significantly. The conclusions of the work became more substantiated. Most of my comments have been corrected.
Reviewer 5 Report
The authors have successfully addressed all comments. The manuscript can be accepted in its present form.